# Influence of Mineralogical Structure of Mold Flux Film on Heat Transfer in Mold during Continuous Casting of Peritectic Steel

**DOI:** 10.3390/ma15092980

**Published:** 2022-04-20

**Authors:** Lei Liu, Xiuli Han, Mingduo Li, Di Zhang

**Affiliations:** College of Mining Engineering, North China University of Science and Technology, Tangshan 063210, China; heutliulei@163.com (L.L.); limingduo@ncst.edu.cn (M.L.); ncstdearz@163.com (D.Z.)

**Keywords:** crystallization ratio, microstructure, flux film, heat transfer, longitudinal crack, peritectic steels

## Abstract

The mineralogical structure of flux films is a critical factor in controlling heat transfer in the mold and avoiding the longitudinal cracking of slabs during the continuous casting of peritectic steel. In this study, the layered structure, crystallization ratio, mineralogical species, and morphology features of flux films were characterized by optical microscopy, X-ray diffraction, and electron-probe microanalysis. Microstructural observation revealed that the normal flux films for peritectic steels present a multilayered structure and high crystallization ratio (60~90 vol%), mainly composed of well-developed crystalline akermanite and cuspidine. In contrast, the films with outstanding flux characteristics with abundant longitudinal cracks on the slab surface have a low crystallization ratio (<50 vol%) or vast crystallite content (>80 vol%). Furthermore, heat transfer analysis showed that the low crystallization ratio and the vast crystallite content of flux films worsen the heat transfer rate or uniformity in the mold, whereas the appropriate thickness and cuspidine content of flux films can improve the heat transfer performance. From the above results, it is concluded that using strong crystalline flux to obtain the ideal mineral phase structure of flux film is one of the important measures for reducing longitudinal cracks during continuous casting of peritectic steel slabs.

## 1. Introduction

Peritectic steel is widely used in manufacturing high-strength automobile plates and ship boards with excellent properties of thermoplasticity and machinability [1,2]. Because of the peritectic reaction during continuous casting of a peritectic steel slab, volume contraction and thermal stress congestion of the solidification shell always occur, more easily causing nonuniform heat transfer and longitudinal cracks [3,4,5,6]. Thus, avoiding longitudinal cracks during the continuous casting of peritectic steel slabs remains a challenge.

In order to improve the slab quality, a large amount of research has been carried out on steel solidification characteristics, continuous casting process conditions, and continuous casting flux [7,8,9,10]. Whereas it can be difficult to change the steel solidification characteristic and the casting process condition, the mold flux has become a central factor preventing slab quality defects [11,12,13,14,15]. In-depth knowledge regarding the heat transfer mechanism of mold flux films is becoming increasingly essential. Shibata et al. [16,17,18] suggested the average thickness of mold flux films for various steel grades can be 0.5–1.75 mm and the interfacial thermal resistance increases with increasing flux film thickness. Tsutsumi et al. [19,20,21,22] measured the surface roughness of solidified flux films using a stylus surface profiler and analyzed the surface roughness near the mold to correlate and evaluate the air gap formation. Choi et al. [23,24,25,26,27,28] considered the crystallization behavior of mold fluxes to be vital in controlling the overall heat transfer in the mold by means of the mold simulator, differential thermal analysis, hot thermocouple technique, and confocal laser scanning microscope. Furthermore, many researchers found that the crystallization ratio of mold flux is a key factor in controlling the heat transfer in the mold [29,30,31,32]. However, the effect of mineralogical structure on the heat transfer property of flux films has not been investigated through a systematic approach.

In this work, the microstructure characteristics of flux films matched for peritectic steels were investigated by optical microscopy, X-ray diffraction (XRD), and electron-probe microanalysis (EPMA). Then, the effect of mineralogical structure on the heat transfer property of flux films was analyzed. The aim of the present work is to ravel out the relationship between the mineralogical structure of flux films and the longitudinal cracks of slabs, and then to select the suitable mold flux for avoiding the longitudinal cracks during continuous casting of peritectic steel slabs. The results will provide a theoretical basis for optimizing mold flux and improving slab quality.

## 2. Experimental Section

### 2.1. Materials

Continuous casting experiments were carried out at Hebei Iron and Steel Group Co., Ltd. in Shijiazhuang, China. Samples of flux films (see Figure 1) were taken from the meniscus region (see Figure 2).

Combined with the surface quality of the slab, four typical samples of flux films were selected. Sample No. 1 represents flux films with normal slab quality for peritectic steel A, and sample No. 2 represents flux films with longitudinal cracks on the surface of the slab for peritectic steel A. It was the same with sample No. 3 and sample No. 4 for peritectic steel B. Some parameters of the typical flux films and the corresponding slab quality are listed in Table 1. Chemical compositions and physical properties of mold fluxes for peritectic steels A and B are listed in Table 2 and Table 3.

### 2.2. Methods

In this work, some samples of flux films from the same group were glued together by epoxy adhesive, and their section along the thickness direction was glued to the slide, ground to a thickness of 0.03 mm, and eventually made into a polished thin section. In addition, other samples of flux films and mold fluxes were completely ground to 0.074 mm before being used.

The thickness, mineralogical composition, crystallization ratio, and microstructure of flux films were analyzed using the polished thin section with a polarizing microscope (Axioskop A1 pol, Carl Zeiss Co., Ltd., Bangkok, Thailand). Then, using the polished thin section with an electron-probe microanalyzer (JXA-8230, JEOL Co., Ltd., Tokyo, Japan), the chemical compositions of different mineral crystals in flux films were confirmed.

The crystal phase composition of powder-like flux films was detected with an X-ray diffractometer (BDX-3200, Bruker Co., Ltd., Seongnam-si, Korea) using Cu-Ka radiation within the scanning range of 10° to 80°. The melting temperature and melting speed of powder-like mold fluxes were measured using a melting point and melting speed tester (RDS-04, Northeastern University in China).

The heat flow density (1673 K) and viscosity (1573 K) of mold fluxes were measured using a mold heat flux simulation and viscosity tester (HF-201, Chongqing University in China). The experimental procedure was as follows: First, 350 g prepared powder-like flux was put in a graphite crucible and heated to 1673 K with the MoSi_2_ furnace of the tester. Then, the water-cooled sensor was immersed into the liquid slag and measured 10 data points of heat flow density within 45 s. Then, the sensor was taken out from the liquid slag, and the flux film adhered to the sensor was obtained (probe diagram of mold heat flow simulator can be seen in Figure 3). After a period of time, when the temperature dropped to 1573 K, a standard rotating spindle was immersed into liquid slag and measured 10 data points of viscosity.

To reduce the error, three or more samples were prepared for each mold flux, and the physical properties of each flux were determined from the average value of the measured data of samples; the difference among the measured data was no greater than 3%.

## 3. Results and Discussion

### 3.1. Mineralogical Composition and Crystallization Ratio of Flux Films

As the formation of flux films in a mold is affected by complex factors, the composition, content, morphology, and size of the crystalline minerals are obviously different, showing the variant heat transfer capacity. When the heat transfer capacity of flux films is poor, it will often lead to longitudinal cracks during casting crack-sensitive steel. In this work, mineralogical compositions, proportions, and crystallization ratio of flux films for the peritectic steels were quantitatively analyzed by using a polarizing microscope.

The polarizing microscope analysis results (Table 4 and Figure 4) show that the main crystalline minerals of flux films for steel A are akermanite, cuspidine, and wollastonite. The crystallization ratio of flux films for steel A varies widely. The crystallization ratio of flux films with good slab quality reaches 85~90%, and the content of akermanite is as high as 75~80%. The glass phase of flux films with longitudinal cracks on the slab surface increases obviously, and their crystallization ratio is only 45~50%.

In contrast, the main crystalline minerals of flux films for steel B are akermanite, cuspidine, and a small account of wollastonite. The crystallization ratio of flux films with good slab quality is as low as 60~65%, but the content of cuspidine is as high as 40~45%. Though the crystallization ratio of flux films with longitudinal cracks on the slab surface is as high as 95%, the content of well-shaped cuspidine is only 12~17%, and the remaining content is akermanite crystallites.

According to the backscattered electron (BSE) micrographs and energy-dispersive spectrometer (EDS) images from the electron-probe microanalyzer (Figure 5 and Figure 6), it can be found that a large number of granular crystals and interlaced crystals are melilite, and a small number of spearhead-shaped crystals are cuspidine. Melilite has many isomorphisms such as akermanite (2CaO•MgO•2SiO_2_) and gehlenite (2CaO•Al_2_O_3_•SiO_2_). Their optical properties are similar, so it is difficult to distinguish them by microscope. However, it can be seen from the XRD analysis results (Figure 7) that the melilite in the flux film mainly is akermanite (2CaO•MgO•2SiO_2_).

The crystallization ability of flux films directly affects the heat transfer in the mold. The flux film with low crystallization has low roughness, which makes thermal conductivity become high, and low precipitation of minerals with higher melting points decreases the interfacial thermal resistance, which may lead to surface longitudinal cracks in the peritectic steel. The mineralogical structure of the flux film determines the quality and production efficiency of the slab. The crystallization ratio of the flux film No. 2 for steel A with longitudinal cracks on the slab surface is obviously lower, worsening the heat transfer uniformity in mold, which is one of the causes of longitudinal cracks of slabs. However, the cuspidine content of the flux film No. 4 for steel B with longitudinal cracks on the slab surface is only 12~17%, which may bring about an excessive heat transfer rate causing longitudinal cracks to occur on the surface of the slab. Moreover, the vast akermanite crystallite content in the flux film No. 4 can cause heat transfer nonuniformity in the mold.

### 3.2. Microstructure of Flux Films

The layered structures, mineralogical species, and morphology features of flux films were analyzed and determined using polarizing microscopy. The layered structure of normal flux film for steel A is obvious, and the crystallizing layer accounts for a large area of the flux film (Figure 8a). The normal flux film of steel B exhibits a typical two-layered structure, and part of the flux film has a multilayered structure (Figure 8c). Compared with the glass layer near the mold side of the two normal flux films, the flux film of steel A has a higher degree of discontinuity than the flux film of steel B. The layered structure of flux film for steel A with longitudinal cracks is a typical three-layered structure with glass–crystalline–glass layers (Figure 8b). The crystallizing layer of flux film for steel B with longitudinal cracks accounts for all the flux film (Figure 8d).

The research results show that the flux films have differences in microstructure, especially in the mineral morphology and particle size. The akermanite and wollastonite in the crystal layer of the flux films are distributed alternately on the shell side, which has an obvious boundary with the cuspidine. The akermanite of the flux films for steel A is more prone to be intertexture-shaped near the shell side and to have a radial and chrysanthemum shape on the mold side. However, the akermanite of flux films for steel B is particle-shaped crystallites with small grain size, and the coarse granular and spearhead-shaped cuspidine of the flux films is concentrated on the mold side.

### 3.3. Heat Transfer Property of Flux Films

The heat flux density can be defined as the amount of heat that passes through the flux film per unit cross-sectional area and per unit time. In this study, the heat flow density of mold fluxes was measured by using the mold heat flux simulator when the temperature was maintained at 1673 K, and the measurements represent maximum and minimum heat transfer and heat transfer uniformity. The heat flow density of mold fluxes for peritectic steels was measured at different times by the simulator, as shown in Figure 9. The maximum heat flow density represents the heat transfer capacity of flux films at the meniscus, and the average heat flow density reflects the heat transfer capacity of flux films in the middle and upper part of the mold. The heat transfer test revealed that the heat flow density decreases gradually as the cooling water pass time is increased. The maximum and average heat flow densities of flux films for steel A are higher than those of flux films for steel B, and the same law applies to the heat transfer property for these films.

### 3.4. Discussion

The relationship between the heat flow density and the mineralogical structure of flux films is presented in Figure 10. It can be seen that the greater the thickness of the slag film is, the lower the hindrance to the heat flux is. The greater the crystalline phase of flux film, the less the obstruction to the heat flow. Furthermore, the high crystallization of the flux film promotes an increase in its solidification thickness. However, early studies [33,34] on heat transfer ignored the influence of the mineralogical composition and proportion of flux films in the casting of peritectic steels. Instead, these studies considered that more resistance to heat transfer arises from the higher crystallization ratio or thickness of flux films. Thus, they were unable to explain the phenomenon of heat flow density dropping and becoming smaller than expected at the lower crystallization ratio and film thickness. As shown in Figure 10, the thickness and the heat flow density of flux films for steel A are higher than those of flux films for steel B. According to the thermal conductivity of minerals, cuspidine has the lowest thermal conductivity among all the mineral phases of flux films. With respect to the mineralogical structure of flux films for steel B, the content of cuspidine is as high as 40~45%, which can increase the thermal resistance of the flux film and maintain the suitable heat transfer rate and uniformity in mold all the time. In addition, there is an exception in Figure 10: the heat flow density drops and becomes smaller at the 90% crystallization ratio of the flux film; it seems perfectly natural to ascribe this to the bigger thickness and higher crystallization ratio of the flux film with the lower impact of cuspidine content. Therefore, considering the effect of the mineralogical structure of flux films on heat transfer in the mold and slab quality, this study carried out a comprehensive work and found that the mineralogical compositions and proportions can also be a key factor in controlling heat transfer suitable for the slab quality, so it is different from previous investigators’ observations which incorrectly ascribed this effect to the crystallization ratio or thickness of flux films.

Flux films of peritectic steel with longitudinal cracks generally have lower crystallization ratios and crystal growth levels than the normal flux films. The direct cause of vast longitudinal cracks on the peritectic steels is the abnormal microstructure of flux films, which may be caused by the inappropriate chemical components of flux and the cooling condition of the mold. The crystallization ratio of the unqualified films for steel A with longitudinal cracks may be only 45~50%. Compared with the reasonable flux films for steel B, the unqualified films have a mass of akermanite crystallite and the crystallization can abnormally reach 80% or more. All of these results suggest that the primary reasons for longitudinal cracking on the slab surface of the peritectic steels are the low crystallization ratio and the vast crystallite content, which can decrease the thermal resistance of flux films and worsen the heat transfer rate and uniformity.

In view of the strong crack sensitivity of peritectic steel, the longitudinal crack ratio of the slabs can be reduced by improving the mineralogical structure of the flux films. However, if the heat transfer is controlled only by enhancing the crystallization capacity, the balance between the heat transfer and lubrication properties of the flux films may be lost. So, during continuous casting of peritectic steels, the contradiction between heat transfer and lubrication should be considered comprehensively, and the optimum mineralogical structure of flux films should be obtained through reasonable composition regulation of the mold flux. In this work, it is recommended that the content of (CaO/SiO_2_) or (Na_2_O + K_2_O) of the mold flux for steel A should be properly increased. While maintaining high alkalinity (CaO/SiO_2_), the CaF_2_ content of the mold flux for steel B should be increased appropriately.

## 4. Conclusions

The influence of the mineralogical structure of mold flux films on heat transfer in the mold during the continuous casting of peritectic steels for avoiding the longitudinal cracks of slabs has been investigated. The following conclusions can be drawn:(1)The mineralogical structure of normal flux films during the continuous casting of peritectic steels presents a multilayered structure with a high crystallization ratio (60~90 vol%), mainly composed of well-developed crystalline akermanite and cuspidine.(2)Flux films for peritectic steels with longitudinal cracks have a lower crystallization ratio and crystal growth level than the normal flux films have, showing the characteristic of low crystallization ratio (<50 vol%) or vast crystallite content (>80 vol%).(3)The results from heat transfer analysis confirmed that the mineralogical composition and proportion of flux films is a key factor in controlling heat transfer during the casting of peritectic steels. The more glass phase and crystallite flux films have, the worse the heat transfer rate and uniformity in the mold are.(4)Using strong crystalline flux to obtain an ideal mineral phase structure of flux films is one of the important measures for reducing the longitudinal cracking of peritectic steels, and the ideal mineral phase is characterized by a high crystallization ratio (>60 vol%), without the vast crystallite content, and as much cuspidine content as possible.

## Figures and Tables

**Figure 1 materials-15-02980-f001:**
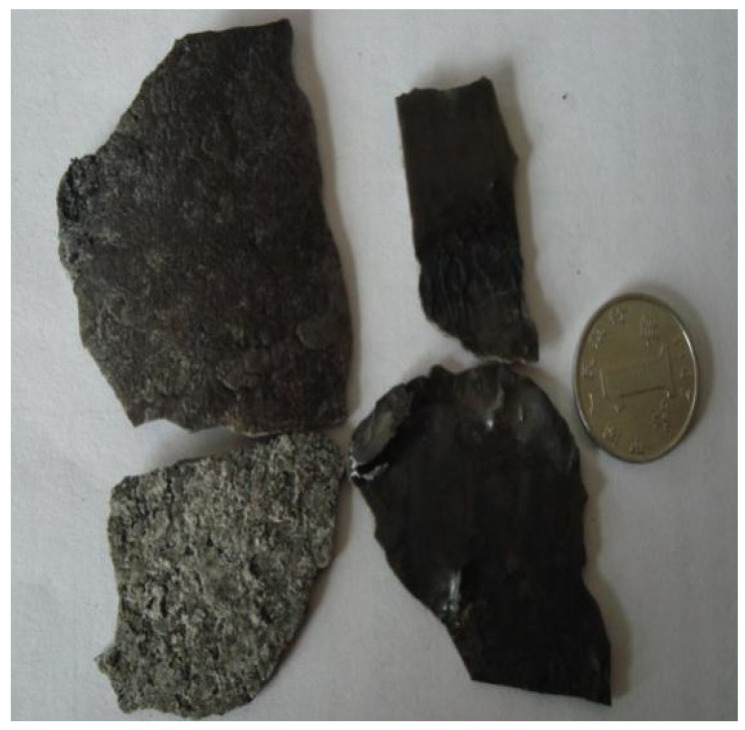
Samples of solidified flux films.

**Figure 2 materials-15-02980-f002:**
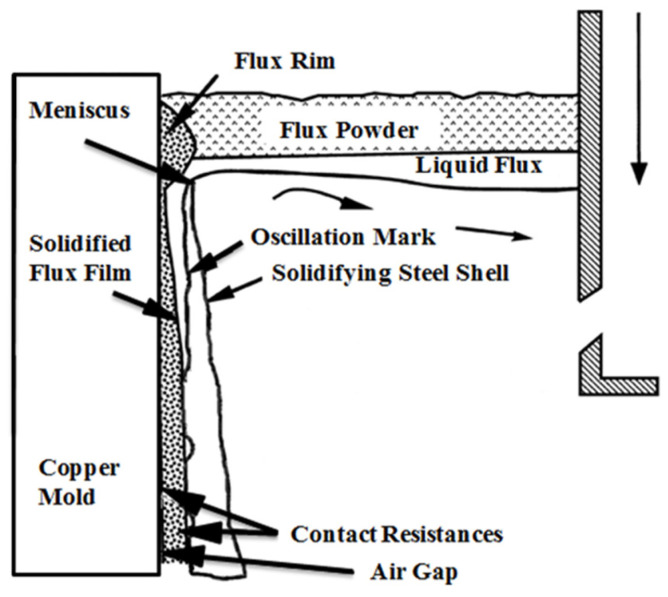
Position diagram of flux films in mold.

**Figure 3 materials-15-02980-f003:**
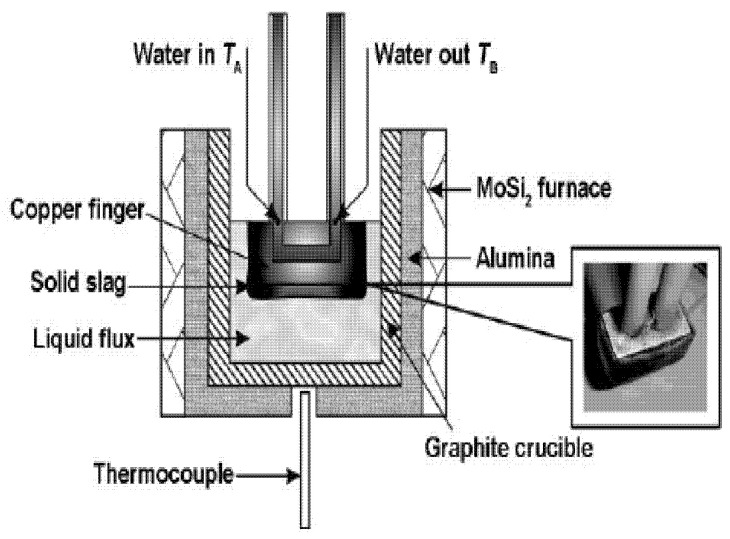
The probe diagram of the mold heat flow simulator.

**Figure 4 materials-15-02980-f004:**
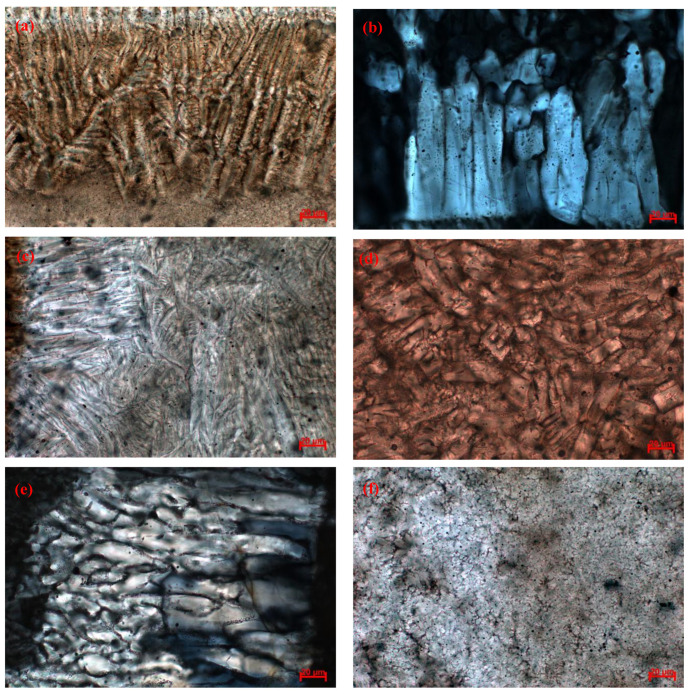
Mineralogical structures in flux films: (**a**) akermanite in film 1; (**b**) cuspidine in film 1; (**c**) akermanite in film 2; (**d**) akermanite in film 3; (**e**) cuspidine in film 3; (**f**) akermanite in film 4.

**Figure 5 materials-15-02980-f005:**
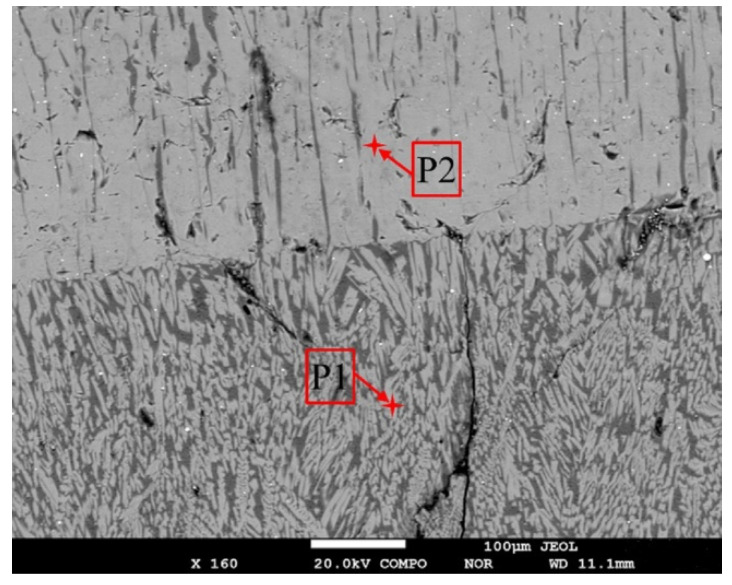
BSE micrographs of flux films by EPMA.

**Figure 6 materials-15-02980-f006:**
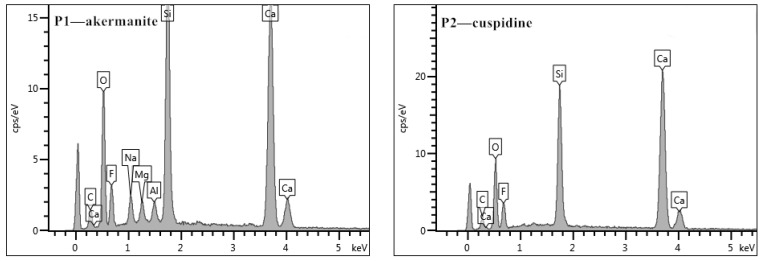
EDS spectra of different mineral crystals in flux films by EPMA.

**Figure 7 materials-15-02980-f007:**
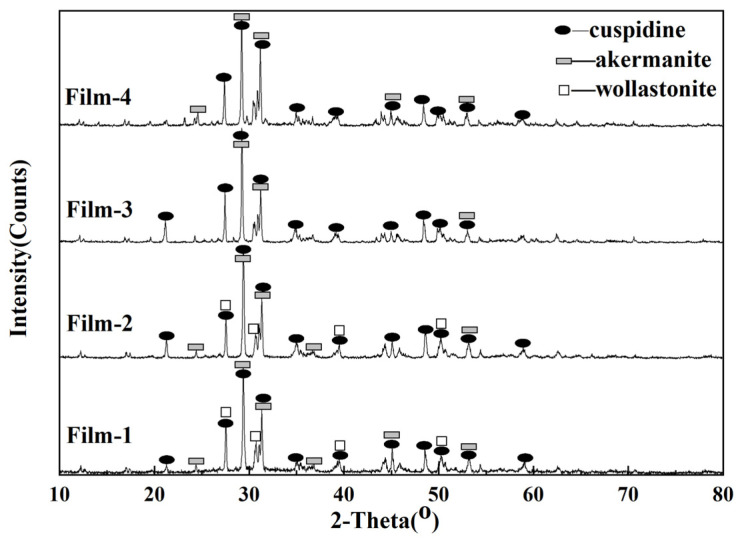
XRD results of flux films.

**Figure 8 materials-15-02980-f008:**
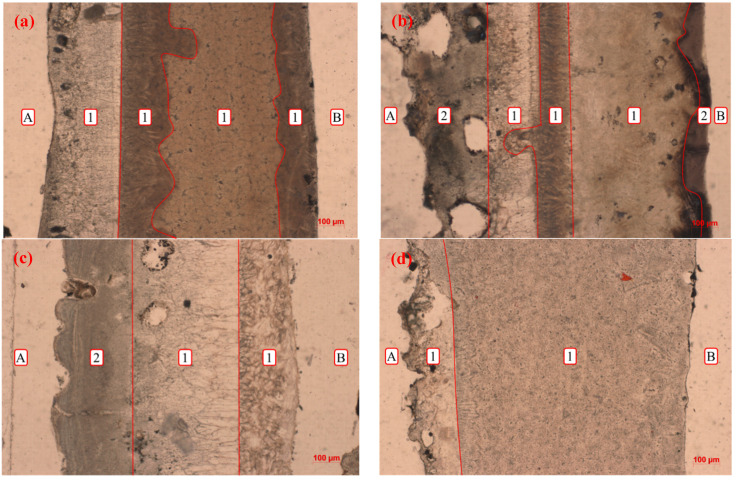
Layered structures of flux films: (**a**) film 1; (**b**) film 2; (**c**) film 3; (**d**) film 4; A—the mold side; B—the shell side; 1—crystallization layer; 2—glass layer.

**Figure 9 materials-15-02980-f009:**
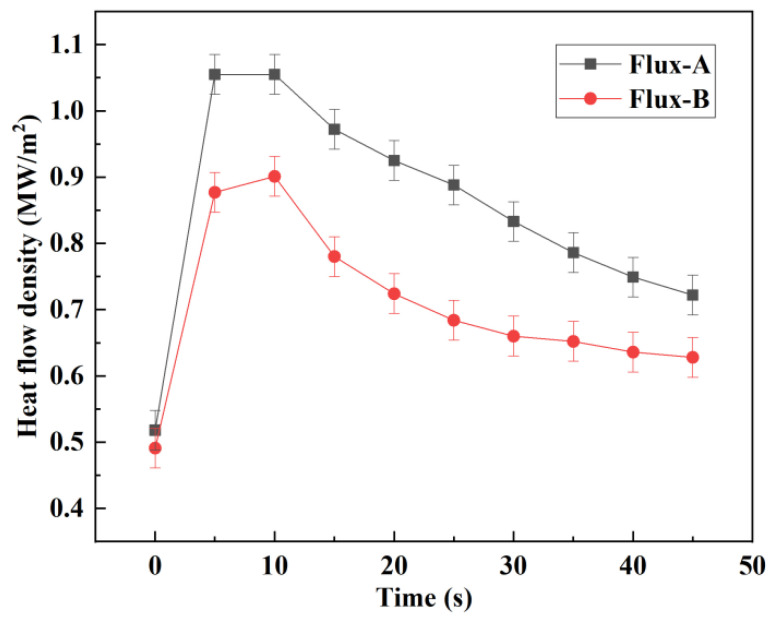
The heat flux density of mold flux films at different times.

**Figure 10 materials-15-02980-f010:**
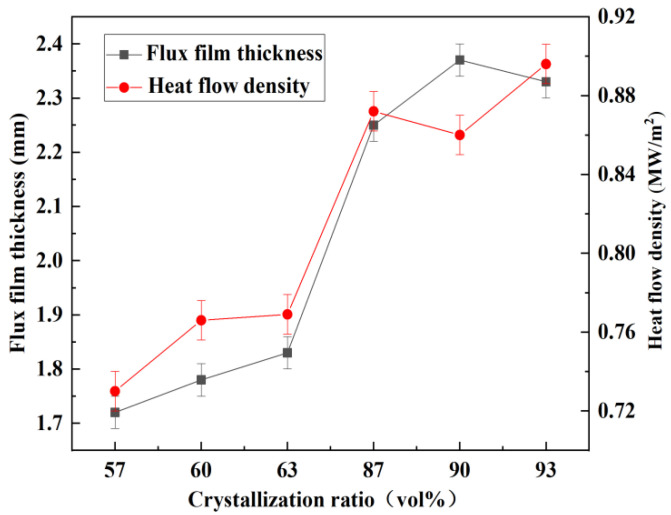
Relationship between the heat flow density and mineralogical structure of flux films.

**Table 1 materials-15-02980-t001:** Parameters of mold flux films for peritectic steels.

Peritectic Steel Number	Flux Number	Film Number	Thickness of Liquid Flux (mm)	Consumption of Mold Flux (kg·t^−1^)	Slab Quality
A	F-A	No. 1	10.25	0.596	Normal
		No. 2	10.41	0.568	Longitudinal crack
B	F-B	No. 3	10.33	0.582	Normal
		No. 4	10.32	0.560	Longitudinal crack

**Table 2 materials-15-02980-t002:** Chemical compositions of mold fluxes (wt%).

Peritectic Steel Number	Flux Number	SiO_2_	Al_2_O_3_	MgO	CaO	Fe_2_O_3_	K_2_O + Na_2_O	MnO	F^−^	C	H_2_O
A	F-A	28.99	2.74	4.17	35.83	0.57	7.65	1.73	7.46	4.88	0.33
B	F-B	31.77	4.18	3.72	36.29	1.03	7.82	1.80	6.87	6.31	0.34

**Table 3 materials-15-02980-t003:** Physical properties of mold fluxes.

Peritectic Steel Number	Flux Number	Alkalinity	Melting Point (°C)	Melting Speed (s)	Viscosity (Pa·s)	Molten Heavy (g/cm^3^)
A	F-A	1.24	1050	21	0.108	0.80
B	F-B	1.14	1104	21	0.153	0.70

**Table 4 materials-15-02980-t004:** Mineralogical compositions, proportions, and crystallization ratio of flux films (vol%).

Peritectic Steel Number	Film Number	Slab Quality	Cuspidine	Akermanite	Wollastonite	Glass Phase	Crystallization Ratio
A	No. 1	Normal	10~15	75~80	1~5	5~10	85~90
	No. 2	Longitudinal crack	25~30	20~25	1~5	45~50	45~50
B	No. 3	Normal	40~45	20~25	<1	30~35	60~65
	No. 4	Longitudinal crack	12~17	80~85	<1	1~5	>95

## Data Availability

Not applicable.

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
