# Peer review of "Influence of Mineralogical Structure of Mold Flux Film on Heat Transfer in Mold during Continuous Casting of Peritectic Steel"

_materials, 2022, doi:10.3390/ma15092980_

Round 1

Reviewer 1 Report

The manuscript is focused on the investigation of mineralogical structure of mold flux film effect on the heat transfer in mold under conditions of continuous casting of peritectic steel. The topic is of high importance in material science. The manuscript is well-presented and logically written. The conclusions are supported by experimental data. There are several remarks to be considered prior to the acceptance.

1. Introduction, several relevant references are missed and have to be added. 

https://doi.org/10.1016/j.jmrt.2019.03.005 https://doi.org/10.1002/srin.202100165 https://doi.org/10.2355/isijinternational.52.1310 2. Abbreviations need revision. The majority of them are nor explained at the first mention. 3. Secion 2.2, the information regarding apparatus producers is fully missed. 4. Fig. 7 is too small and almost unreadable. 5. Fig. 9 and 10, each point should be presented with corresponding error bar. Otherwise, a single measurement results are shown that is inappropriate. The manuscript can be accepted to publication after minor revision.

Reviewer 2 Report

Dear Authors,

Please find comments on the paper “Influence of mineralogical structure of mold flux film on heat transfer in mold during continuous casting of peritectic steel”.

The article describes the effects of the mineralogical structure of mold fluxes on heat transfer during continuous casting of peritectic steels. The research performed in this work fits the scope of Materials, and it would be of interest to the readers due to the significant industrial application of peritectic steels. However, the manuscript does not show the technical rigor necessary to be published in this journal. Thus, major corrections are suggested before the manuscript is considered for publication in Materials.

Suggestions:

Improve technical details of the techniques, equipment and parameters used for the following melting points, viscosity, mold fluxes thickness testing performed. The current information is insufficient for clarity to our readers.

Figure 2 looks very similar to published images in the literature; please draw a new figure or get the proper license.

In figure 3 indicate the location of the mold fluxes for heat transfer measurements in the probe diagram of the mold heat flow simulator.

In figure 4, please improve the scale labels in the images; they are too small to be visible.

Figure 6, the footnote should be EDS spectra of different mineral crystals in flux films by EPMA

Please improve the quality of XRD data in figure 7 since it is almost impossible to observe the identified phases.

The authors are encouraged to emphasize what is new compared with other works in the field in their discussion.

Reviewer 3 Report

Please emphasize the novelty of this study.
Figures 9 and 10 have no error bars.
Please add more detailed information on methods. How "polished thin section with polarizing microscope (Axioskop A1 pol), combining X-ray diffractometer (BDX-3200) and electron probe microanalyzer (JXA-8230)" carried out? Has it been combined together in a customized test stand or used separately? If separately, how long was the delay between measurements? On the other hand, if used together how was it combined into one test stand? As a reader it is hard for me to figure out how the experiment was carried out. Please add more details.
Please add discussion or literature review with  ISIJ International, Vol. 38 (1 998), No. 8, pp. 834-842 and ISIJ International. Vol. 38 (1 998), No. 5, pp. 440-446, where results and observations of heat transfer and thermal resistance of mold flux film in process of continuous mold casting is also analyzed experimentally and numerically. 

Reviewer 4 Report

This study focused on the influence of mineralogical structure of mold flux film on heat transfer in mold during a continuous casting of peritectic steel in order to diminish a chance of slab longitudinal crack. The present investigation is useful in academic viewpoint as well as for applications related to casting of peritectic steel. However, there are some unclear and missing points as follows that need to be reasonably addressed before it may be publishable in Materials J.

  1. Concerning the measured data, the authors should show that the heat flow density and viscosity data are statistically meaningful through uncertainty analysis and repeatability of data.
  2. What does it mean by heat transfer uniformity? Does it mean temperature distribution? If so, how were temperature at various locations measured? And where were the measured locations?
  3. In figure 9, to make it clearer, in place of Flux-1 and Flux-2, they should be Flux-A and Flux-B respectively.
  4. In the Discussion section, referring to figure 10, it says in the manuscript that “the greater the thickness of the slag film is, the greater the hindrance to the heat flux is”. However, it looks to me from the plot that the greater the thickness of the slag film is, the lower the hindrance to the heat flux is as the heat flow density increases with the film thickness.
  5. In figure 10, what could have happened to the value of heat flow density that drops and becomes smaller than expected at the 90 % crystallization rate?

Round 2

Reviewer 2 Report

Dear authors,

Please find comments on corrections performed by the authors on the paper “Influence of mineralogical structure of mold flux film on heat transfer in mold during continuous casting of peritectic steel”.

As mentioned before, the research of this work fits the scope of Materials, and it would be of interest to the readers due to the significant industrial application of peritectic steels.

The authors made the corrections suggested, which are found reasonable but with some minor spelling details in the 2.2 section. Once the spelling details of the 2.2 section have been addressed, the manuscript can be published in Materials.

Reviewer 4 Report

My comments have been reasonably addressed. The manuscript were correctly modified. The manuscript has been greatly improved. Therefore I recommend the paper for publication in the Materials J.

Author Response

    Thank you very much for your approval to the corrections of our manuscript. The authors would like to take this opportunity to express our appreciation for your constructive comments and suggestions to improve the quality of our paper. Best wishes to you.